# AdaRD-key: Adaptive Relevance-Diversity Keyframe Sampling for Long-form Video understanding

## Abstract

Understanding long-form videos remains a significant challenge for vision-language models (VLMs) due to their extensive temporal length and high information density. Most current multimodal large language models (MLLMs) rely on uniform sampling, which often overlooks critical moments, leading to incorrect responses to queries. In parallel, many keyframe selection approaches impose rigid temporal spacing: once a frame is chosen, an exclusion window suppresses adjacent timestamps to reduce redundancy. While effective at limiting overlap, this strategy frequently misses short, fine-grained cues occurring near important events. Other methods instead emphasize visual diversity, but neglect to consider query relevance. We propose **AdaRD-Key**, a training-free keyframe sampling module for query-driven long-form video understanding. AdaRD-Key maximizes a unified Relevance–Diversity Max-Volume (RD-MV) objective, which combines a query-conditioned relevance score with a log-determinant diversity component to yield informative yet non-redundant frames. To handle broad queries with weak alignment to the video, AdaRD-Key employs a lightweight relevance-aware gating mechanism. When the relevance distribution indicates weak alignment with the video, the method seamlessly shifts into a diversity-only mode, thereby enhancing coverage without requiring additional supervision. Our entire pipeline is training-free, computationally efficient (running in real time on a single GPU), and compatible with existing VLMs in a plug-and-play manner. Extensive experiments on LongVideoBench and Video-MME further demonstrate that AdaRD-Key achieves state-of-the-art performance, particularly on long-form videos.

## 1 Introduction

With the rapid progress of multimodal large language models (MLLMs) (Zhang et al., 2024a; Wang et al., 2024a; Zhang et al., 2024d; Li et al., 2024a), several high-performing video understanding systems, such as Video-ChatGPT (Maaz et al., 2023) and Video-LLaMA (Zhang et al., 2023), have emerged, and the research focus is gradually shifting toward long-video understanding (Weng et al., 2024; Shu et al., 2025; Wang et al., 2024c). Most existing models (Li et al., 2024c; Liao et al., 2024; Shen et al., 2024; Ren et al., 2024; Weng et al., 2024; Jin et al., 2024), however, still rely on uniform sampling of frames (Kim et al., 2024; Wang et al., 2024d;c). The longer the video, the sparser the sampled information becomes, making it increasingly likely that the model will miss query-relevant content. Current key-frame selection methods fall mainly into three categories, each with clear limitations: **i) Redundancy-only Maximization.** These methods (Li et al., 2025) select frames that are temporally or visually diverse, often by maximizing dissimilarity over time. However, because they ignore query relevance, they may select frames that are visually distinct yet semantically irrelevant; **ii) Relevance plus recursive bisection.** Some methods (Tang et al., 2025) integrate query relevance with a recursive interval-splitting strategy to achieve coverage across the video timeline. While this improves coverage, these methods do not explicitly enforce semantic diversity, and they struggle with densely packed or temporally adjacent relevant events; **iii) Training-based query-frame association.** Other techniques (Cheng et al., 2025; Hu et al., 2025; Buch et al., 2025; Liang et al., 2024; Tan et al., 2024) learn a dedicated model to predict query-relevant keyframes. Although this improves alignment with the query, it introduces a high training burden and still fails to eliminate near-duplicate frames due to local score clustering.

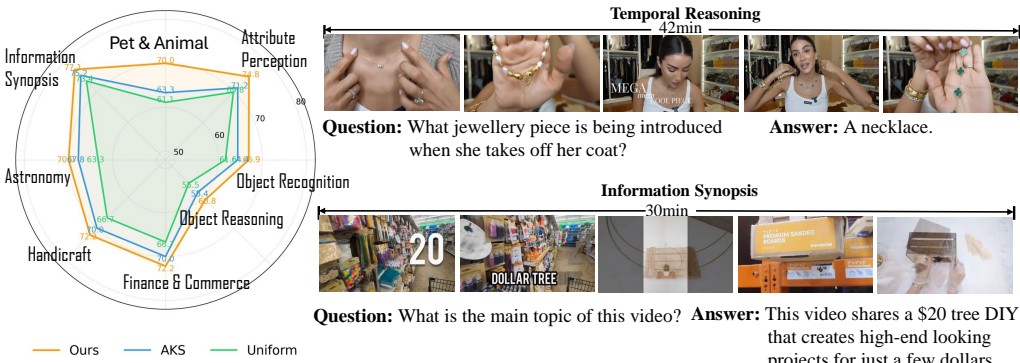

Figure 1: Video-MME results. Left: Category-wise accuracy: AdaRD-Key (ours) forms the outer envelope vs. Uniform and AKS (larger radius is better). Right: Two long-video examples (Temporal Reasoning, 42 min; Information Synopsis, 32 min) where our selected keyframes capture the needed evidence and yield correct answers.

To address these limitations, we propose AdaRD-Key, a training-free, query-driven keyframe sampling module that directly balances query relevance and visual diversity for long-video understanding. We target two objectives: (1) **Relevance**-Identify frames that are highly relevant to the given query; (2) **Diversity**-Preserve diversity among those frames to avoid redundancy and improve evidence coverage. We first compute frame–query similarity using BLIP-2 (Li et al., 2023) and cache the frame-level features. Next, we apply our proposed **Relevance–Diversity Max-Volume (RD-MV)** selector- a greedy objective that maximizes relevance while jointly enforcing a log-determinant-based diversity constraint to suppress redundant frames. To further adapt to query characteristics, we introduce **Variability–Budget Scaling (VB-Scale)**, which adjusts the relevance–diversity trade-off based on the score variability and budget size. When the query–frame relevance distribution is weak or diffuse (e.g., in abstract queries), AdaRD-Key employs a relevance-aware gating mechanism, which dynamically shifts the selector into a diversity-only mode—guided by the same max-volume criterion—to recover broad coverage without additional supervision. This performance aligns with our design principles: the RD-MV objective jointly optimizes for relevance and de-redundancy, while VB-Scale adaptively reallocates selection budget to ensure balanced coverage across diverse semantic phases. In long videos (e.g., 600 seconds), our approach significantly improves evidence retention without compromising efficiency. Figure 1 demonstrates the superiority of AdaRD-Key over state-of-the-art methods, highlighting its effectiveness in overcoming the limitations of previous approaches. The left radar compares Uniform, AKS (current SoTA), and AdaRD-Key across Videomme sub-categories (e.g., Finance & Commerce, Pet & Animal, Handicraft, Technology, Documentary). AdaRD-Key traces the outer envelope on most axes, indicating consistent gains over both AKS and Uniform. In the right panel, for a 42-min temporal reasoning example, AdaRD-Key captures coherent long-range cues and produces the correct answer; in the 32-min information synopsis example, it similarly aggregates diverse evidence to summarize the main topic. Together, these results underscore AdaRD-Key's strength in coupling relevance with de-redundancy over long video horizons.

The entire pipeline is training-free, real-time capable on a single GPU, and compatible with any vision-language model without the need for fine-tuning or architectural modifications. Our contributions are threefold: **i) Joint Relevance–Diversity Objective.** AdaRD-Key introduces RD-MV, the first keyframe sampling method to jointly optimize query relevance and embedding-space diversity for long-video understanding, bridging the gap between redundancy-focused and relevance-only methods. **ii) Variability–Budget Scaling (VB-Scale).** We introduce a training-free mechanism that automatically balances relevance and diversity based on the signal variability and the available frame budget. A lightweight relevance-aware gate further stabilizes selection when query–video alignment is weak. **iii) Plug-and-Play Deployment.** AdaRD-Key is completely training-free and model-agnostic. It can be applied directly to any off-the-shelf VLM on a single A100 GPU, facilitating seamless integration across datasets, domains, and tasks.

## 2 RELATED WORK

**Long-form Video Understanding.** Recent work (Wu et al., 2024b; Korbar et al., 2024) in long video understanding tackles the challenge of reasoning over sequences spanning several minutes

to hours—durations that far exceed the token budgets of conventional short-clip VLMs. For example, Hierarchical Differential Distillation (HDD) (Cheng et al., 2025) distills teacher models into compact multi-resolution representations, keeping salient content while aggressively reducing spatial-temporal tokens. Memory-augmented transformers (He et al., 2024) such as ViT-Mem style architectures cache and reuse visual features across successive segments so the model need not re-encode the entire stream each time. To capture dependencies that span large temporal ranges, Temporal Perceiver style models query learned latent slots over extended videos, whereas state-space or other linear-time sequence formulations (e.g., VideoMamba) (Li et al., 2024b) replace full quadratic self-attention with scalable temporal kernels that maintain context efficiently. To support these advances, new long-video benchmarks and datasets include LongVALE (Geng et al., 2025), LongVideoBench (Wu et al., 2024a) with extended-context adapters (e.g., VideoLLaMA2 (Cheng et al., 2024)), and grounding corpora such as GroundVQA (Di & Xie, 2024) Ego4D NLQ (Grauman et al., 2022), and MomentSeeker (Yuan et al., 2025). Collectively, these approaches typically rely on compressing, memorizing, and selectively attending to sparse signals, which inevitably leads to substantial information loss and often retains only a small fraction of truly useful evidence.

**Keyframe Selection for Long Video Understanding.** Recent keyframe or clip-selection methods (Park et al., 2024; Guo et al., 2025) for long-video understanding can be grouped into several representative approaches. Adaptive Keyframe Sampling (AKS) (Tang et al., 2025) formulates selection as maximizing Relevance + Temporal Coverage under a fixed budget and solves it with a judge-and-split recursion; however, its coverage term measures only time-axis uniformity, so substantial semantic redundancy may persist within the chosen frames. The M-LLM-based selector queries a frozen multimodal LLM twice—first on single frames, then on caption sequences—to re-rank relevance, but the two-stage inference is computationally heavy and requires supervised tuning. Flexible Frame Selection (FFS) (Buch et al., 2025) introduces a differentiable gating module that learns to decide, per question, how many frames to keep, balancing accuracy and cost; yet the method focuses on frame-count elasticity and leaves inter-frame redundancy largely untreated. Finally, MaxInfo (Li et al., 2025) is a training-free heuristic that selects frames maximizing the log-determinant (volume) of their embedding matrix, offering speed and parameter-free deployment, but being query-agnostic and unaware of temporal structure. In summary, existing methods overlook redundancy reduction for diversity, whereas our work introduces a simple yet effective algorithm for video understanding that preserves relevance while promoting diversity.

# 3 METHOD

## 3.1 PRELIMINARIES

Long videos inherently contain a vast amount of information; however, only a small portion of this content is typically relevant to answering a given query. The central challenge, therefore, lies in accurately and comprehensively identifying query-relevant information within an extended video sequence. We formalize the input video as $V \in \mathbb{R}^{N \times W \times H \times C}$, where $N$ is the total number of frames, and $W$, $H$, and $C$ represent the frame width, height, and channels, respectively. To extract frame-level semantic features, we employ BLIP-2 (Li et al., 2023), which encodes each frame with respect to the query and produces a set of frame embeddings $\{f_1, f_2, ..., f_N\}$, along with corresponding relevance scores $R(f)$ indicating alignment with the query. Our goal is to select a subset $\widehat{F} \subseteq \{f_1, f_2, ..., f_k\}$ of $k$ keyframes that maximizes a joint relevance–diversity objective, defined as:

$$\widehat{F} \;=\; \arg\max_{F, |F|=k} \Big[ \underbrace{\sum_{f \in F} R(f)}_{\text{Relevance}} + \lambda \underbrace{D(F)}_{\text{Diversity}} \Big],$$

(1)

Here, the first term $\sum_{f \in F} R(f)$ promotes the selection of frames that are highly relevant to the query. The second term $D(F)$ measures the diversity of the selected frames and is defined as:

$$G_F \;=\; E_F^{\top} E_F, \qquad \mathcal{D}(F) \;=\; \log \det\big(G_F + \varepsilon I\big)$$

(2)

In this formulation, $E_F$ is the matrix formed by stacking the feature vectors of the selected frames in $\widehat{F}$, and $G_F$ is the resulting Gram matrix capturing pairwise similarities. The additive term $\varepsilon I$ ensures numerical stability and guarantees that the logarithm is well-defined. The diversity term $D(F)$ is a monotone submodular function (Krause & Golovin, 2014), which satisfies diminishing

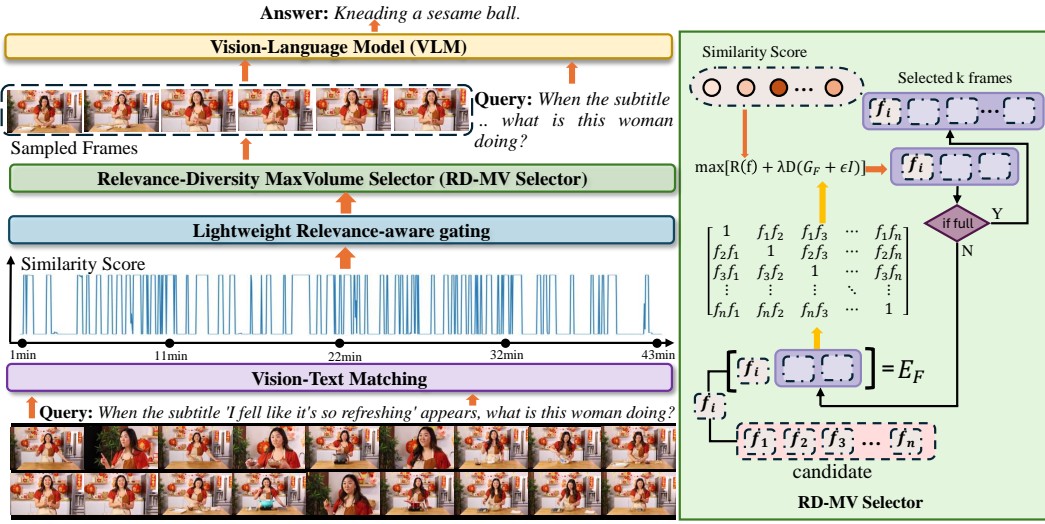

Figure 2: Overview of the proposed framework. Video frames are sampled and scored against the query using BLIP-2. A lightweight relevance-aware gate determines whether to adopt a relevance–diversity or diversity-only strategy. The RD-MV selector then greedily selects keyframes to maximize $R(f) + \lambda D(F)$, which are fed into the VLM for downstream tasks.

returns and enables efficient optimization. Due to the submodular structure of $\log \det(\cdot)$, we can approximate the solution to this objective using a greedy Max-Volume algorithm, which selects frames that offer the highest marginal gain in each iteration. This ensures that Equation 1 can be efficiently maximized in practice, balancing both query relevance and semantic diversity within the selected frame subset.

## 3.2 ARCHITECTURE.

We propose AdaRD-Key, a training-free, query-driven keyframe selector for long-video understanding. As shown in Figure 2, given a video (sampled at 1 fps by default) and a text query, we first compute frame–query relevance scores and cache L2-normalized visual embeddings using a BLIP-2-based encoder. Keyframes are then chosen by our Relevance–Diversity Max-Volume (RD-MV) selector, which greedily maximizes a joint objective that sums relevance with a log-determinant diversity term to suppress near-duplicate frames; an efficient Sherman–Morrison update keeps the procedure linear in the number of selected frames. To adapt this trade-off across videos of different lengths and score profiles, we introduce Variability–Budget Scaling (VB-Scale), which sets the diversity weight from two interpretable signals: the variability of the relevance scores (how "peaky" they are) and the budget ratio $\rho = N/K$ (candidate frames per slot, where $N$ denotes the total number of frames at 1 fps and $K$ is the target number of keyframes). In addition, a lightweight relevance-aware gate monitors alignment (e.g., max-score/entropy heuristics) and, for weakly aligned queries, falls back to a diversity-only selection to avoid amplifying noise. The entire pipeline is plug-and-play (no training, no fine-tuning), runs efficiently on a single GPU, and returns $K$ temporally ordered keyframes that cover complementary evidence while staying focused on the query.

### 3.2.1 LIGHTWEIGHT RELEVANCE-AWARE GATE

The Lightweight Relevance-Aware Gate plays a critical role in determining whether the computed query–frame relevance scores are reliable enough to guide selection. In cases where the query is under-specified (e.g., "describe the scene") or only weakly aligned with the video, the resulting relevance scores are often uniformly low and noisy. When the query is strongly correlated with the frames, the objective is Relevance + Diversity. Conversely, when the correlation is weak, the objective switches to Diversity. By switching to the Diversity-only mode, sampling becomes more varied and covers a broader range of information.

Accordingly, we let $R_i \in [0, 1]$ denote the relevance score of frame $i$, computed as the image-text matching probability from BLIP-2 Li et al. (2023), evaluated at a default sampling rate of 1 frame

per second (fps). We define a binary alignment indicator $\eta$ based on the maximum relevance score $R_{max}$:

$$\eta = \begin{cases} 1, & max(R) \geq \tau \\ 0, & \text{else} \end{cases} \tag{3}$$

where the threshold $\tau$ is set to 0.4, if $\eta = 1$ (i.e., alignment is strong), we proceed with the original joint relevance–diversity objective described in Equation 1. However, if $\eta = 0$ (i.e., the query exhibits weak alignment with the video), we suppress the relevance term and switch to a diversity-only objective, optimizing:

$$\max_{\mathcal{F}:|\mathcal{F}|=K} \log \det(\boldsymbol{I} + \boldsymbol{E}_{\mathcal{F}} \boldsymbol{E}_{\mathcal{F}}^{\mathsf{T}}), \tag{4}$$

14This fallback ensures that, without strong query grounding, the selected keyframes still achieve broad coverage and low redundancy by maximizing the spanned volume in the embedding space.

### 3.2.2 RELEVANCE-DIVERSITY MAXVOLUME SELECTOR(RD-MV SELECTOR)

We begin by extracting frame-level features and query-conditioned relevance scores using BLIP-2 Li et al. (2023), Based on these, our Relevance–Diversity Max-Volume (RD-MV) Selector chooses a set of K keyframes from the candidate pool. In contrast to prior keyframe selection approaches, our method explicitly optimizes for two properties: **(1) Relevance,** ensuring that each selected frame is closely aligned with the input query, and **(2) Diversity,** promoting mutual dissimilarity among frames to minimize redundancy. The joint objective is defined in Equation 1, which we maximize via a greedy strategy, detailed below.

As shown in the right panel of Figure 2, the relevance term $R(f)$ is additive across the selected set. The diversity term $D(F)$ is computed as the log-determinant of the similarity matrix formed by the selected frame features, which can be interpreted geometrically as the volume of the spanned subspace. A higher volume indicates lower inner-product correlation between frame embeddings, thus encouraging the selection of diverse content. Importantly, $D(F)$ is a monotone submodular function, and since the sum of two monotone submodular functions is also monotone submodular Krause & Golovin (2014), the overall objective retains this property. This allows us to efficiently maximize it with a greedy algorithm, which iteratively selects the feature vector $f_i$ with the greatest marginal gain $\Delta(i|F)$. Let the selected set $F$ contain keyframes $E_F = [f_1, f_2, ...f_k] \in \mathbb{R}^{d \times k}$, and let $G_F \in \mathbb{R}^{k \times k}$ denote the Gram matrix of the selected features. For a candidate frame $i$ with normalized feature vector $f_i \in \mathbb{R}^d$, we define $r = E_F^T f_i \in \mathbb{R}^{k \times 1}$ as the vector of inner products between the candidate and the selected set. If $f_i$ is appended to $E_F$, the updated similarity matrix becomes $G_{F \cup \{i\}}$:

$$G_{F \cup \{i\}} = \begin{bmatrix} G_F & r \\ r^{\mathsf{T}} & 1 \end{bmatrix} \tag{5}$$

According to the Equation 5, the volume of $G_{F \cup \{i\}}$ can be computed as:

$$\det(G_{F \cup \{i\}} + \varepsilon I) = \det(G_F + \varepsilon I) \left(1 - r^{\mathsf{T}} G_F^{-1} r + \varepsilon\right) \tag{6}$$

From this, the diversity gain of adding frame $i$ can be calculated as:

$$\underbrace{\log \det(G_{F \cup \{i\}} + \varepsilon I) - \log \det(G_F + \varepsilon I)}_{\Delta \text{Diversity}} = \log((1 + \varepsilon) - \mathbf{r}^{\mathsf{T}} (G_F + \varepsilon I)^{-1} \mathbf{r}) \tag{7}$$

The total marginal gain for candidate frame $i$ is therefore:

$$\Delta(i \mid F) = R(i) + \lambda \log((1 + \varepsilon) - \mathbf{r}^{\mathsf{T}} (G_F + \varepsilon I)^{-1} \mathbf{r}) \tag{8}$$

At each iteration, the algorithm selects the candidate frame with the highest marginal gain:

$$i^* = \arg\max_{i \in \mathcal{U} \setminus F} \Delta(i \mid F), \quad F \leftarrow F \cup \{i^*\}, \tag{9}$$

At each iteration, the algorithm selects the candidate frame with the highest marginal gain: Since the diversity gain in Equation 7 depends only on the current Gram matrix $(G_F + \varepsilon I)^{-1}$, we can maintain

and update it efficiently using the Sherman–Morrison formula, avoiding repeated full-matrix inversions. Details on efficiently updating the inverse of $(G_F + \varepsilon I)^{-1}$ are provided in Equations 21– 24 of the Appendix.

### 3.2.3 VARIABILITY-BUDGET SCALING (VB-SCALE)

In the RD-MV selector, a single parameter $\lambda$ controls the trade-off between query relevance and diversity. Selecting an appropriate value for $\lambda$ s essential for effective keyframe selection. We propose VB-Scale, a lightweight, training-free mechanism that adaptively sets $\lambda$ using two key signals: **1) Video length**, expressed as the budget ratio $\rho = T/K$, where $T$ is the number of candidate frames and $K$ is the selection budget. A higher $\rho$ (i.e., more frames per selection slot) implies that stronger diversity pressure is needed, so $\lambda$ should increase. **2) Relevance score variability.** which reflects how peaked or flat the query–frame relevance distribution is. When the distribution is sharp (i.e., already "focused"), the model can rely more on relevance and less on diversity. When it is flat or diffuse, diversity becomes more important.

**Variability-Driven Component:** To quantify score variability, we compute the coefficient of variation (CV) of the relevance scores $R$:

$$CV = \frac{\text{std}(R)}{\text{mean}(R) + \epsilon}. \tag{10}$$

Accordingly, we define a decreasing function of CV as a variability-sensitive diversity weight:

$$\lambda_{\text{var}} = \lambda_{\text{min}} + \frac{\lambda_{\text{max}}}{1 + \alpha \cdot CV}. \tag{11}$$

where $\lambda_{min}$ and $\lambda_{max}$ are lower and upper bounds for $\lambda$, and $\alpha$ controls sensitivity to $CV$. As CV decreases (i.e., relevance becomes flatter), $\lambda_{var} \to \lambda_{min} + \lambda_{max}$, increasing diversity pressure. When $CV$ is high (i.e., relevance is peaky), $\lambda_{var} \downarrow \lambda_{min}$, allowing relevance to dominate.

**Budget-Driven Component:** For the length-based signal, we define the budget ratio $\rho = T/K$. This expresses how many candidate frames are available per slot. We map $\rho$ to the range $[0, \lambda_{max}]$ using a saturating logarithmic function:

$$\lambda_{\text{bud}} = \lambda_{\text{max}} \min\left(1, \frac{\log(\rho + \epsilon)}{\log(\rho_{\text{cap}})}\right). \tag{12}$$

Here, $\rho_{cap} = 8$ defines the saturation point beyond which increases in $\rho$ no longer raise $\lambda$. This prevents runaway growth of the diversity term in extremely long videos.

When $\rho$ is small near budget, and $\lambda_{bud}$ is small for diversity pressure. As $\rho$ grows, $\lambda_{bud}$ rises and saturates at $\lambda_{max}$, preventing runaway growth. where $\rho_{cap} = 8$ is the saturation point set for $\rho$. This prevents the diversity weight from increasing indefinitely as the video length continues to grow.

Finally, blend the two with a logistic weight on $\rho$:

$$w(\rho) = \frac{1}{1 + \exp(-(\rho - \rho_0))} \in (0, 1). \tag{13}$$

The final diversity weight $\lambda$ is computed as:

$$\lambda = \text{clip}(w\,\lambda_{\text{bud}} + (1 - w)\,\lambda_{\text{var}}, \lambda_{\text{min}}, \lambda_{\text{max}}). \tag{14}$$

We use the following settings throughout our experiments: $\lambda_{min} = 0.05$, $\lambda_{max} = 0.6$, $\alpha = 2.0$, $\rho = 8$, $\rho_0 = 1$.

Our pipeline is training-free, single-pass, and plug-and-play. We cache frame features once, compute query–frame relevance, and select $K$ keyframes via the RD-MV objective with VB-Scale and a lightweight relevance-aware gate ($\eta$) for weak alignment cases. The selector's runtime scales roughly linearly with the number of candidate frames $T$ with a small $m \leq K$ factor due to Sherman-Morrison updates, and memory footprint is dominated by storing $N \times d$ features at 1-fps. In practice we use $K \in \{32, 64\}$, temperature-free BLIP-ITM scores (no fine-tuning), VB-Scale with

Figure 3: Qualitative results on LongVideoBench using 64 keyframes. Blue text highlights scene constraints; red text marks critical evidence. AdaRD-Key (Ours) captures key details missed by Uniform and AKS, leading to correct answers with LLaVA-Video.

$\lambda \in [\lambda_{min}, \lambda_{max}]$, and a simple gate threshold on $max(R)$ to fall back to diversity-only when alignment is weak. This yields a deterministic, efficient, and tunable selection mechanism that prioritizes query-relevant and non-redundant frames, improving downstream vision-language model (VLM) performance on long-video reasoning tasks—without any additional training. The complete algorithmic procedure of AdaRD-Key is provided in Algorithm 2 of the appendix.

# 4 EXPERIMENTS

## 4.1 EXPERIMENTAL SETUP AND DETAILS

**Dataset and evaluation.** We evaluate AdaRD-Key using the LMMS-Eval (Zhang et al., 2024b) framework across two widely used benchmarks for long-form video understanding: LongVideoBench and VideoMME. These datasets include videos that can span over an hour, making high-quality keyframe selection essential. We integrate AdaRD-Key into two prominent multimodal large language models (MLLMs), namely LLaVA-Video and Qwen2-VL (Wang et al., 2024b), without fine-tuning or modifying any of their parameters. This setup ensures that the observed performance gains are solely attributed to the quality of selected keyframes. To emphasize the role of visual input and minimize reliance on language priors, we exclude subtitles during evaluation, thereby isolating the models' visual reasoning capabilities.

**Implementation details.** For implementation, we sample video content at 1 frame per second, producing a candidate pool of $N$ frames. Using BLIP-2 (Li et al., 2023), we extract visual features ft and compute frame–query relevance scores. AdaRD-Key then selects the final $K$ keyframes using its RD-MV objective, guided by the Variability–Budget Scaling (VB-Scale) mechanism and the lightweight relevance-aware gate. The pipeline is training-free, operates on a single A100 (80GB) GPU, and is evaluated with Qwen2-VL (Wang et al., 2024b), and LLaVA-Video (Zhang et al., 2024c). We report results using 32 frames for Qwen2-VL (Wang et al., 2024b) and 64 frames for LLaVA-Video. For each question, MLLMs are prompted with a standardized multiple-choice format conditioned on the query and the selected visual inputs. Code will be released on GitHub.

## 4.2 COMPARISON TO THE STATE-OF-THE-ART

**Quantitative results.** We first report performance on LongVideoBench, as shown in Table 1, where our method is compared to several recent baselines. With 32 keyframes, AdaRD-Key raises the accuracy of Qwen2-VL to 60.8%, a notable improvement over the vanilla Qwen2-VL and surpassing the state-of-the-art M-LL Selector by 3.8 percentage points. While the gain over AKS is modest at 0.3 points, our method consistently performs better across longer video durations. With 64 keyframes, AdaRD-Key achieves 62.9%, improving upon MAXINFO by 1.4 points and AKS by 0.2 points. Although it slightly underperforms AKS by 0.3% in the 15–60 minute video category, it delivers a 1.3% improvement over AKS in the 3–10 minute category, indicating superior performance in shorter-to-medium-length scenarios. On VideoMME (as shown in Table 2), AdaRD-Key also demonstrates robust gains. Applied to Qwen2-VL with 32 frames, it lifts the overall score to 60.7%, outperforming the vanilla Qwen2-VL baseline by 3.1 points, Q-Frame by 2.4 points, and AKS by 0.8 points. Performance improvements are more pronounced for longer video segments: for medium-length videos, accuracy reaches 59.2%, reflecting a 4.0-point gain over vanilla, 2.1 over Q-Frame, and 1.5 over AKS. For long videos, the score rises to 51.9%, surpassing vanilla by 4.5

Table 1: LongVideoBench (LVB val) accuracy of different models and keyframe selection methods. Bold indicates models with AdaRD-Key, which consistently improves performance over all training-free baselines.

| Method | Frames | LLM | Training free | overall | (8,15] | (15,60] | (180,600] | (900,3600] |
|---|---|---|---|---|---|---|---|---|
| *GPT-4o (OpenAI, 2024)* | 64 | – | ✗ | 62.0 | 71.4 | 76.7 | 61.4 | 55.8 |
| *Gemini-1.5-Flash (Team et al., 2023)* | 64 | – | ✗ | 56.8 | 68.3 | 76.2 | 54.4 | 48.6 |
| *Gemini-1.5-Pro (Team et al., 2023)* | 64 | – | ✗ | 58.6 | 67.4 | 75.1 | 59.3 | 50.9 |
| VideoLLaVA (Lin et al., 2023) | 8 | 7B | ✗ | 39.1 | 43.1 | 44.6 | 36.4 | 34.4 |
| IDEFICS2 | 32 | – | ✗ | 47.8 | 59.8 | 65.7 | 47.8 | 42.7 |
| PLLava (Xu et al., 2024) | 32 | 34B | ✗ | 53.2 | 60.1 | 66.8 | 50.8 | 49.1 |
| MANTIS-IDEFICS2 | 32 | – | ✗ | 45.4 | 56.6 | 55.8 | 42.7 | 40.4 |
| Qwen2-VL (Wang et al., 2024b) | 32 | 7B | ✗ | 55.5 | 70.4 | 69.8 | 53.1 | 46.8 |
| Qwen2-VL w/ M-LLM sector (Hu et al., 2025) | 32 | 7B | ✗ | 57.0 | – | – | – | – |
| Qwen2-VL w/ AKS (Tang et al., 2025) | 32 | 7B | ✓ | 60.5 | 70.4 | 69.8 | 59.2 | 54.3 |
| **Qwen2-VL w/ AdaRD-Key** | 32 | 7B | ✓ | **60.8** | 70.4 | 69.8 | **60.4** | **55.0** |
| LLaVA-Video (Zhang et al., 2024c) | 64 | 7B | ✗ | 58.9 | 72.0 | 72.1 | 58.3 | 51.2 |
| LLaVA-Video/ MAXINFO (Li et al., 2025) | 64 | 7B | ✓ | 61.5 | – | – | – | – |
| LLaVA-Video w/ AKS (Tang et al., 2025) | 64 | 7B | ✓ | 62.7 | 72.0 | 72.1 | 61.6 | **57.4** |
| **LLaVA-Video w/ AdaRD-Key** | 64 | 7B | ✓ | **62.9** | 72.0 | 72.1 | **62.9** | 57.1 |

points, Q-Frame by 3.6, and AKS by 0.8. Even in shorter videos, where gains are typically harder to achieve, AdaRD-Key delivers a slight improvement with 71.1%, edging out all three baselines.

Table 2: Accuracy on Video-MME (V-MME). Bold rows denote models with AdaRD-Key. Short videos refer to content under 2 minutes, medium videos range from 4–15 minutes, and long videos span 30–60 minutes.

| Method | Frames | LLM | Training free | Overall | Short | Medium | Long |
|---|---|---|---|---|---|---|---|
| VideoLLaVA (Lin et al., 2023) | 8 | 7B | ✗ | 37.6 | 42.7 | 37.1 | 33.0 |
| InternVL2 (Chen et al., 2024) | 8 | 8B | ✗ | 52.6 | 62.4 | 51.1 | 44.2 |
| LongVU (Shen et al., 2024) | – | 7B | ✗ | 60.9 | 64.7 | 58.2 | 59.5 |
| Frame-Voyager (Yu et al., 2024) | – | 8B | ✗ | 57.5 | 67.3 | 56.3 | 48.9 |
| Qwen2-VL (Wang et al., 2024b) | 32 | 7B | ✗ | 57.6 | 70.8 | 55.2 | 47.4 |
| Qwen2-VL w/ Q-Frame (Zhang et al., 2025a) | 44 | 7B | ✗ | 58.3 | 69.4 | 57.1 | 48.3 |
| Qwen2-VL w/ AKS (Tang et al., 2025) | 32 | 7B | ✓ | 59.9 | 70.4 | 57.7 | 51.1 |
| **Qwen2-VL w/ AdaRD-Key** | 32 | 7B | ✓ | **60.7** | **71.1** | **59.2** | **51.9** |
| LLaVA-Video (Zhang et al., 2024c) | 64 | 7B | ✗ | 64.4 | 76.6 | 62.3 | 54.4 |
| LLaVA-Video/ MAXINFO (Li et al., 2025) | 64 | 7B | ✓ | 64.2 | 74.6 | 63.3 | 54.6 |
| LLaVA-Video w/ AKS (Tang et al., 2025) | 64 | 7B | ✓ | 65.3 | **77.1** | 64.7 | 53.9 |
| **LLaVA-Video w/ AdaRD-Key** | 64 | 7B | ✓ | **65.6** | 76.7 | **64.9** | **55.3** |

**Qualitative Results.** Figure 3 demonstrates the comparative performance of Uniform sampling, Adaptive Keyframe Selection (AKS), and our proposed AdaRD-Key. In each query, blue text indicates scene or background constraints, while red text highlights the critical information required to answer the question. In the left panel of Figure 4, Uniform sampling fails to retrieve frames containing the essential content. AKS, influenced by dominant scene or background features, overlooks the crucial detail—the "40 cm" reference from the query. In contrast, AdaRD-Key, which integrates both Relevance and Diversity signals, successfully selects frames that directly correspond to the query's focus. A similar trend is evident in the right panel of Figure 4, where AdaRD-Key again outperforms the other methods by aligning its keyframe selection more closely with the query intent. The VideoMME benchmark results with 32-frame inputs to Qwen2-VL, comparing Uniform, AKS, and AdaRD-Key, are illustrated in Figures 4 and 5 of the appendix.

Table 3: Video captioning results on VCapsBench (Zhang et al., 2025b) using 4-frame sampling. Qwen-2.5VL-7B (Team, 2025) is the baseline; w/ AdaRD-Key applies our keyframe-aware module.

| Model | Size | Frames | AR↑ | IR↓ | CR↑ |
|---|---|---|---|---|---|
| Qwen-2.5VL | 7B | 4 | 44.86 | 28.26 | 62.54 |
| **Qwen-2.5VL w/ AdaRD-Key** | 7B | 4 | **52.41** | **19.11** | **64.79** |

**Video Captioning** We conduct experiments not only on long-video VQA, but also on video captioning. For the latter, we adopt the Video Caption Evaluation Benchmark (VCapsBench (Zhang et al., 2025b)). Because videos in this benchmark range from 4 to 34 seconds, we sample four frames per video for our experiments. We use Qwen-2.5-VL-7B as the baseline model and Gemini-2.5-Flash-Lite as the evaluator. Our evaluation metrics are Accuracy (AR), Inconsistency Rate (IR), and Coverage Rate (CR).

As shown in Table 3, beyond VQA, our method also works well for video captioning. Qwen-2.5VL performs solidly with accuracy (AR) of 44.68, inconsistency rate (IR, lower is better) of 28.26, and coverage rate (CR) of 62.54. After inserting our module, AR rises to 52.41 (+7.55), IR decreases by 9.51, and CR increases to 64.79 (+2.25). Overall, these results indicate that our approach substantially benefits video captioning. The quantitative results for video captioning are presented in Figure 6.

### 4.3 ABLATIVE STUDIES

AdaRD-Key offers an efficient solution for video understanding by integrating four components: Relevance, Diversity, lightweight relevance-aware gating (LRG), and Variability-Budget Scaling (VB-Scale). To assess the contribution of each, we conduct experiments under five configurations: Uniform sampling, Relevance alone, Relevance + Diversity, Relevance + Diversity + LRG, and the full pipeline with Relevance + Diversity + LRG + VB-Scale. These configurations help isolate the impact of each module.

Table 4: Combined ablation on **LongVideoBench** (LLaVA-Video, 64 keyframes) and **Video-MME** (Qwen2-VL, 32 keyframes)

| Sampling | LongVideoBench | | | Video-MME | | |
|---|---|---|---|---|---|---|
| | Overall | (180, 600] | (900, 3600] | Short | Medium | Long |
| Uniform | 58.9 | 58.3 | 51.2 | 70.8 | 55.2 | 47.4 |
| Top( Relevance ) | 62.3 | 61.7 | 56.4 | 70.7 | 55.3 | 50.9 |
| Relevance + Diversity | 61.6 | 60.9 | 55.5 | **72.0** | 58.4 | 51.6 |
| R+D + VB-Scale | _62.7_ | _62.4_ | _56.9_ | 71.4 | _58.6_ | _51.8_ |
| R+D + VB-Scale+ LRG (AdaRD-Key) | **62.9** | **62.9** | **57.1** | _71.1_ | **59.2** | **51.9** |

Table 4 shows how performance evolves with the incremental addition of modules. It reports experimental results on LongVideoBench with 64-frame inputs using LLaVA-Video, and on Video-MME with 32-frame inputs using Qwen-2VL. Each module leads to consistent performance gains, especially for longer videos. On LongVideoBench, accuracy for 180–600s videos improves from 58.3 to 62.9 (+4.6), and from 51.2 to 57.1 (+5.9) for videos between 15 minutes and 1 hour, yielding a +4.0 overall increase. Similarly, for Video-MME, gains are observed across all video lengths: +0.3 for short, +4.0 for medium, and +4.5 for long videos. For LRG, although the method is relatively simple, it proves highly effective for abstract tasks such as video summarization. Its performance is illustrated in Figure 7 of the appendix. In summary, results from both LongVideoBench and Video-MME highlight that performance improvements from AdaRD-Key become more pronounced as video duration increases.

## 5 CONCLUSION

We introduced AdaRD-Key, a training-free keyframe selection method for long-video understanding that balances query relevance and visual diversity. Its RD-MV selector jointly optimizes relevance and log-determinant diversity to reduce redundancy, while VB-Scale adapts this balance to video length and score variability. A lightweight relevance-aware gate further improves robustness by reverting to diversity-only mode when query alignment is weak. AdaRD-Key consistently outperforms uniform sampling, top-$k$ relevance, and prior selection strategies on LongVideoBench and Video-MME under both 32- and 64-frame budgets, with notable gains on longer videos. Beyond VQA, it also improves video captioning on VCapsBench under constrained frame budgets. These results show that RD-MV with VB-Scale effectively surfaces salient, diverse content for downstream vision–language models—without training or fine-tuning. The approach is plug-and-play, efficient on a single GPU, and entirely training-free.

ETHICS STATEMENT

This work focuses on developing a training-free keyframe selection method for long-video understanding. All experiments are conducted on publicly available datasets (LongVideoBench, Video-MME, and VCapsBench), which do not involve personally identifiable or sensitive information. No human subjects were recruited or impacted in this study. We have carefully considered issues of fairness, bias, and potential misuse: AdaRD-Key is designed as a generic algorithmic component and does not introduce additional risks beyond those already present in the underlying datasets and vision–language models. The code and data processing steps will be released to ensure transparency and research integrity.

REPRODUCIBILITY STATEMENT

We have taken several measures to ensure the reproducibility of our work. The complete implementation of AdaRD-Key (including the RD-MV selector, VB-Scale mechanism, and relevance-aware gating module) will be released in an anonymous repository, together with scripts for reproducing all reported results. Details of datasets (LongVideoBench, Video-MME, and VCapsBench) and pre-processing steps (frame extraction, keyframe sampling, and evaluation protocols) are described in the supplementary materials. All hyperparameters and their values are explicitly provided in the main text. Additional ablation studies and derivations of the RD-MV objective and block matrix inversion updates are also provided in the appendix to support theoretical claims. With the provided code, dataset references, and detailed descriptions of training-free baselines, all results reported in the paper can be fully reproduced.

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

APPENDIX

USE OF LARGE LANGUAGE MODELS (LLMS)

Large Language Models (LLMs) were used in this work solely as an auxiliary tool for language editing, translation, and LaTeX formatting assistance. They were not used for idea generation, experimental design, data analysis, or drawing conclusions. All research contributions, including the development of AdaRD-Key, theoretical derivations, experimental design, and result interpretation, were entirely carried out by the authors. The authors take full responsibility for the content of this paper. The implementation details of our algorithm are provided in the appendix.

## A METHOD EXTENSION.

First, we provide a detailed derivation of the incremental computation of diversity in our method. In addition, we present a detailed procedure for efficiently updating the inverse matrix $(G_{F \cup \{i\}} + \varepsilon I)^{-1}$ when a new $f_i$ is added.

### A.1 PROOF OF DIVERSITY GAIN

For the gain of diversity, we provide a step-by-step derivation from Equation 5 to Equation 7. Firstly, The Equation 5 is:

$$G_{F \cup \{i\}} = \begin{bmatrix} G_F & r \\ r^\top & 1 \end{bmatrix} \tag{15}$$

For this equation, we first construct a lower triangular matrix as follows:

$$L = \begin{bmatrix} I & 0 \\ -r^\top G^{-1} & 1 \end{bmatrix}, \qquad \det(L) = 1. \tag{16}$$

Then, $LG$ is:

$$LG_{F \cup \{i\}} = \begin{bmatrix} G_F & r \\ 0 & 1 - r^\top G_F^{-1} r \end{bmatrix}. \tag{17}$$

Now $LG_{F \cup \{i\}}$ is an upper block triangular matrix, and its determinant equals the product of the determinants of its diagonal blocks:

$$\det(LG_{F \cup \{i\}}) = \det(G_F) \left( 1 - r^\top G_F^{-1} r \right). \tag{18}$$

Since $det(L) = 1$, it follows that $det(G_{F \cup \{i\}}) = \det(LG_{F \cup \{i\}})$. Hence,

$$\det(G_{F \cup \{i\}} + \varepsilon I) = \det(G_F + \varepsilon I) \left( 1 - r^\top (G_F + \varepsilon I)^{-1} r + \varepsilon \right) \tag{19}$$

Therefore, Equation 19 can be further simplified to Equation 7:

$$\underbrace{\log \det(G_{F \cup \{i\}} + \varepsilon I) - \log \det(G_F + \varepsilon I)}_{\Delta \text{Diversity}}$$
$$= \log[\det(G_F + \varepsilon I) \left( 1 - r^\top G_F^{-1} r + \varepsilon \right)] - \log \det(G_F + \varepsilon I) \tag{20}$$
$$= \log(1 - \mathbf{r}^\top (G_F + \varepsilon I)^{-1} \mathbf{r} + \varepsilon)$$

Equations 15 to 20 provide the complete derivation of the diversity gain, which forms the basis of Equation 7 in the main content.

### A.2 SHERMAN-MORRISON INVERSE MATRIX UPDATE.

As discussed in the main body of the paper, assume that in the current iteration, the keyframe set $F$ has already obtained $m$ keyframes, so the inverse matrix $(G_F + \varepsilon I)^{-1}$ has dimensions $m \times m$.

In the next iteration, after adding a new keyframe $f_i$, the dimensions of $(G_{F \cup \{i\}} + \varepsilon I)^{-1}$ become $(m+1) \times (m+1)$. For the newly constructed matrix $(G_{F \cup \{i\}} + \varepsilon I)^{-1}$, the expression is given by:

$$(G_{F \cup \{i\}} + \varepsilon I) = \begin{bmatrix} G_F + \varepsilon I & r \\ r^\top & 1 + \varepsilon \end{bmatrix} \tag{21}$$

Since $(G_{F \cup \{i\}} + \varepsilon I)$ satisfies the invertibility condition of the corresponding Schur complement, its explicit inverse can be directly derived. Here, we formally introduce the following definition: $A := G_F + \varepsilon I$, $B := A^{-1}$, then compute the Schur complement:

$$\alpha = (1 + \varepsilon) - r^\top A^{-1} r = (1 + \varepsilon) - r^\top B r, \tag{22}$$

Apply the block-inverse formulat(with $A$ invertible)

$$(G_{F \cup \{i\}} + \varepsilon I)^{-1} = \begin{bmatrix} A^{-1} + A^{-1} r \alpha^{-1} r^\top A^{-1} & -A^{-1} r \alpha^{-1} \\ -\alpha^{-1} r^\top A^{-1} & \alpha^{-1} \end{bmatrix}. \tag{23}$$

Let $y := Br = A^{-1} r$ and rewrite:

$$(G_{F \cup \{i\}} + \varepsilon I)^{-1} = \begin{bmatrix} B + yy^\top / \alpha & -y/\alpha \\ -y^\top / \alpha & 1/\alpha \end{bmatrix}. \tag{24}$$

The original $B = (G_F + \varepsilon I)^{-1}$ is expanded to the inverse of the augmented matrix after adding the new frame. The *top-left block* equals $B + yy^\top / \alpha$, and the *new column/row* are $-y/\alpha$ and $1/\alpha$, respectively.

### A.3 DETAILS OF THE AdaRD-KEY ALGORITHM

AdaRD-Key first estimates frame–query relevance and caches $\ell_2$-normalized visual embeddings with a BLIP-based encoder. Keyframes are then selected by our Relevance–Diversity Max-Volume (RD-MV) rule, which greedily maximizes a joint objective that sums relevance with a log-determinant diversity term to suppress near-duplicate frames via a rank-one Sherman–Morrison update that maintains the inverse Gram matrix, keeping each selection step linear in the number of chosen frames; see Algorithm 1 for details. To adapt this trade-off across videos with different lengths and score distributions, we introduce Variability–Budget Scaling (VB-Scale), which sets the diversity weight from two interpretable signals: the variability of the relevance scores (how "peaky" they are) and the budget ratio $\rho = N/K$ (candidate frames per slot, where $N$ is the total number of frames at 1 fps and $K$ is the target number of keyframes). Finally, a lightweight relevance-aware gate monitors alignment (e.g., via max-score or entropy heuristics) and switches to a diversity-only mode for weakly aligned queries to avoid amplifying noise. The full algorithmic details are shown in Algorithm 2.

## B   MORE VISUALIZATION RESULTS

Figures 4 and 5 compare Uniform, AKS, and AdaRD-Key on Qwen2-VL with a 32-frame budget, where orange spans denote ground-truth keyframe regions. In Fig. 4 the answer-bearing content is concentrated in a short window: AdaRD-Key places several selections inside this region (e.g., four hits), while Uniform misses it and AKS captures at most one, leading the baselines to incorrect answers. In Fig. 5 the evidence is distributed over multiple disjoint time segments: Uniform and AKS either cluster around incidental peaks or land off-target, whereas AdaRD-Key retrieves complementary frames from several spans (e.g., three hits). This behavior arises from the RD-MV objective, which adds a log-determinant diversity term to suppress near-duplicates, together with VB-Scale, which sets the diversity weight using both the variability of relevance scores and the frame-budget ratio $\rho = N/K$. When the relevance profile is peaky, VB-Scale tempers diversity so the selector concentrates around the true segment; when $\rho$ is large or the profile is flatter, it increases diversity to cover separated segments. As a result, AdaRD-Key supplies the VLM with the right evidence and achieves the correct prediction, while the baselines fail to collect the necessary cues.

**Caption→QA evaluation.** Figure 6 illustrates our pipeline for assessing the usefulness of selected frames via captioning. For each video–question pair, AdaRD-Key selects a small set of keyframes

---

**Algorithm 1** RD-MV selector: greedy relevance–diversity max-volume

---

**Require:** embeddings $E \in \mathbb{R}^{d \times N}$ (cols $\ell_2$-normalized as $e_i$); effective relevance $R^{\mathrm{eff}} \in \mathbb{R}^N$; target size $k$; diversity weight $\lambda$; stability $\varepsilon > 0$
**Ensure:** index set $F$ with $|F| = k$

1:   $F \leftarrow \varnothing$
2:   $G^{-1} \leftarrow$ empty $0 \times 0$ matrix                            $\triangleright \, (G_F + \varepsilon I)^{-1}$
3:   **while** $|F| < k$ **do**
4:      $bestScore \leftarrow -\infty; \; bestIdx \leftarrow -1$
5:      **for** $i = 1$ **to** $N$ **and** $i \notin F$ **do**
6:         **if** $|F| = 0$ **then**
7:            $q \leftarrow 0$                                $\triangleright$ no selected set yet
8:         **else**
9:            $\mathbf{r} \leftarrow E_{:,F}^\top e_i$                           $\triangleright \, \mathbf{r} \in \mathbb{R}^{|F|}$
10:          $q \leftarrow \mathbf{r}^\top G^{-1} \mathbf{r}$
11:         **end if**
12:         $\Delta \leftarrow R_i^{\mathrm{eff}} + \lambda \log(1 - q + \varepsilon)$
13:         **if** $\Delta > bestScore$ **then**
14:           $bestScore \leftarrow \Delta; \; bestIdx \leftarrow i$
15:         **end if**
16:      **end for**
17:      **if** $|F| = 0$ **then**
18:         $F \leftarrow \{bestIdx\}; \quad G^{-1} \leftarrow \left[ (1 + \varepsilon)^{-1} \right]$
19:      **else**
20:         $\mathbf{r}^* \leftarrow E_{:,F}^\top e_{bestIdx}$                   $\triangleright$ to current $F$
21:         $\alpha \leftarrow (1 + \varepsilon) - \mathbf{r}^{*\top} G^{-1} \mathbf{r}^*$         $\triangleright$ Schur complement
22:         $\mathbf{y} \leftarrow G^{-1} \mathbf{r}^*$
23:         $G^{-1} \leftarrow \begin{bmatrix} G^{-1} + \dfrac{\mathbf{y}\,\mathbf{y}^\top}{\alpha} & -\dfrac{\mathbf{y}}{\alpha} \\[2ex] -\dfrac{\mathbf{y}^\top}{\alpha} & \dfrac{1}{\alpha} \end{bmatrix}$     $\triangleright$ block inverse of $(G_{F \cup \{bestIdx\}} + \varepsilon I)$
24:         $F \leftarrow F \cup \{bestIdx\}$
25:      **end if**
26: **end whilereturn** $F$

---

under a fixed frame budget. Using only these frames, Qwen2.5-VL generates a video-level caption. We then pass the original question together with this caption only (no visual input) to Gemini-2.5-Flash-Lite, which produces an answer. A correct answer indicates that the caption contains sufficient evidence, hence the selected frames provide strong question-relevant coverage. The two examples show that captions derived from AdaRD-Key frames allow the judge to correctly answer queries about scene lighting and object plurality, demonstrating that our sampler captures dispersed, time-varying information and surfaces it in text.

**LVG Result.** Figure 7 illustrates the effect of the Lightweight Relevance-Aware Gate (LRG) under weak query–video alignment. In this example, the query is generic and the BLIP-2 frame-wise relevance estimates are uniformly low and noisy. With the gate enabled ("diversity-only", left), AdaRD-Key ignores the unreliable relevance signal and selects keyframes solely by the diversity term, yielding a temporally dispersed set that broadly covers the video and preserves complementary evidence—leading to the correct answer. Without the gate ("relevance + diversity", right), the selector overfits to spurious peaks in the noisy relevance curve, concentrating keyframes near a few bursts and missing large portions of the content; the answer is consequently incorrect. This qualitative case supports our design choice: when alignment is weak, treating relevance as noise and reverting to diversity-only selection improves temporal coverage and downstream accuracy.

---

**Algorithm 2** AdaRD-Key (overall): VB-Scale + relevance-aware gate + RD-MV selector

---

**Require:** embeddings $E \in \mathbb{R}^{d \times N}$; raw relevance $\hat{R} \in \mathbb{R}^N$; target size $k$; stability $\varepsilon > 0$; gate $\tau$;
   VB-Scale hyperparams $(\lambda_{\min}, \lambda_{\max}, \alpha, \rho_{\text{cap}})$
**Ensure:** index set $F$ with $|F| = k$

1: $e_i \leftarrow \dfrac{E_{:,i}}{\|E_{:,i}\|_2}$ for $i = 1, \ldots, N$             $\triangleright$ $\ell_2$-normalized features

2: $R \leftarrow \hat{R}$             $\triangleright$ no normalization; $R \in [0,1]^N$

3: $\mu \leftarrow \text{mean}(R);\ \ \sigma \leftarrow \text{std}(R);\ \ \text{CV} \leftarrow \dfrac{\sigma}{\mu + \delta}$             $\triangleright$ $\delta > 0$ for stability

4: $T \leftarrow N;\ \ \rho \leftarrow \dfrac{T}{k}$

5: $\lambda_{cv} \leftarrow \lambda_{\min} + \dfrac{\lambda_{\max}}{1 + \alpha \cdot \text{CV}}$

6: $\lambda_{\rho} \leftarrow \lambda_{\max} \cdot \min\!\left(1, \dfrac{\log \rho}{\log \rho_{\text{cap}}}\right)$

7: $w \leftarrow \dfrac{1}{1 + e^{-(\rho-1)}}$

8: $\lambda \leftarrow \text{clip}\!\left(w\,\lambda_\rho + (1-w)\,\lambda_{cv}, \lambda_{\min}, \lambda_{\max}\right)$

9: **if** $\max(R) < \tau$ **then**

10:      $R^{\text{eff}} \leftarrow 0;\ \ \lambda \leftarrow 1$             $\triangleright$ diversity-only fallback

11: **else**

12:      $R^{\text{eff}} \leftarrow R$

13: **end if**

14: $F \leftarrow \text{RD-MV\_SELECT}(E, R^{\text{eff}}, k, \lambda, \varepsilon)$ **return** $F$

---

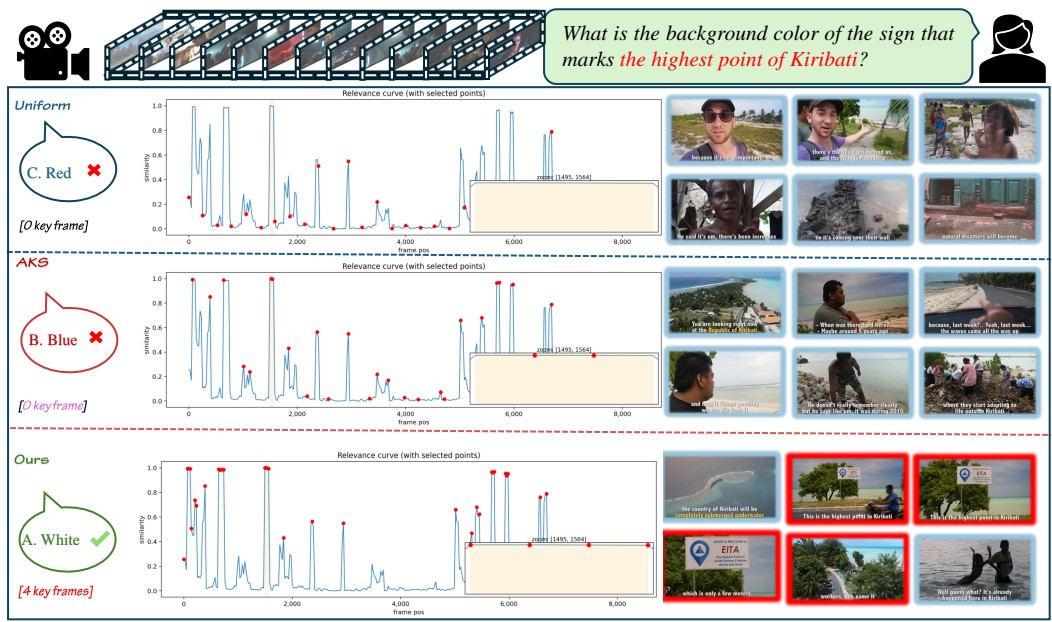

Figure 4: Performance of Qwen2-VL with a $K = 32$ frame budget under three sampling strategies: Uniform, AKS, and AdaRD-Key (ours), for the query shown at the top. For each method, the left speech bubble shows the model's predicted answer option and its correctness (✓/✗) using the frames selected by that sampler. The middle plot shows the relevance curve with selected points (red); orange shaded spans mark ground-truth keyframe regions, and the bracketed number under each row reports how many selected frames fall inside these regions. Thumbnails on the right display the sampled frames; red borders highlight key evidence.

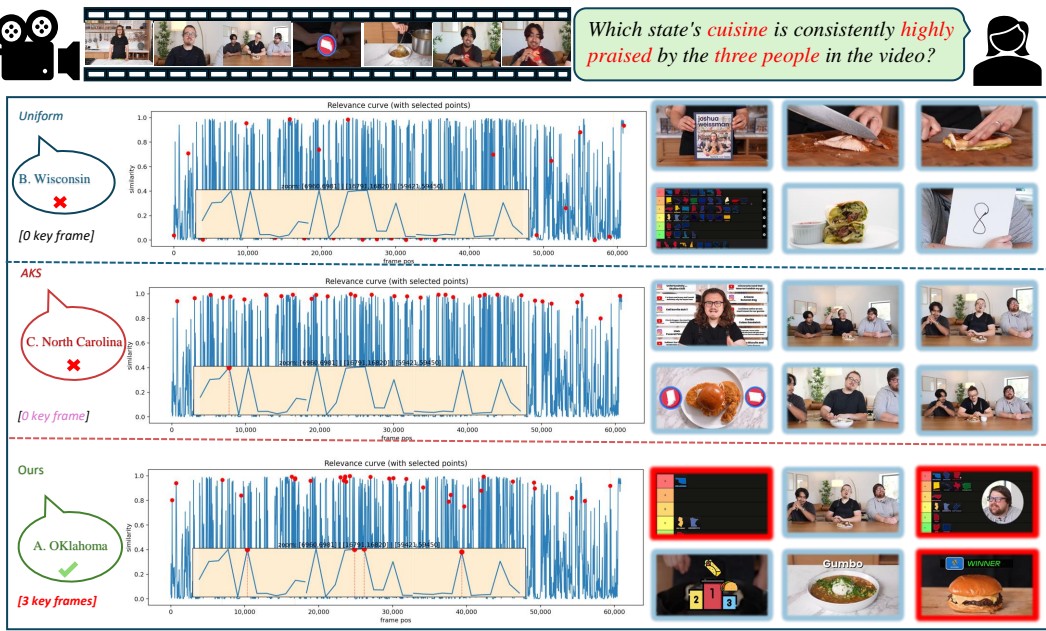

Figure 5: Performance of Qwen2-VL on Video-MME with a $K = 32$ frame budget under three sampling strategies—Uniform, AKS, and AdaRD-Key (ours)—for the query shown at the top. For each method, the left speech bubble reports the model's predicted answer and its correctness (correct/incorrect) using the frames returned by that sampler. The middle plot shows the relevance curve with selected points (red). Orange shaded spans denote the union of multiple *disjoint* ground-truth keyframe regions distributed across the video; the bracketed number under each row counts how many selected frames fall inside any of these spans. Thumbnails on the right depict the sampled frames; red borders highlight key evidence.

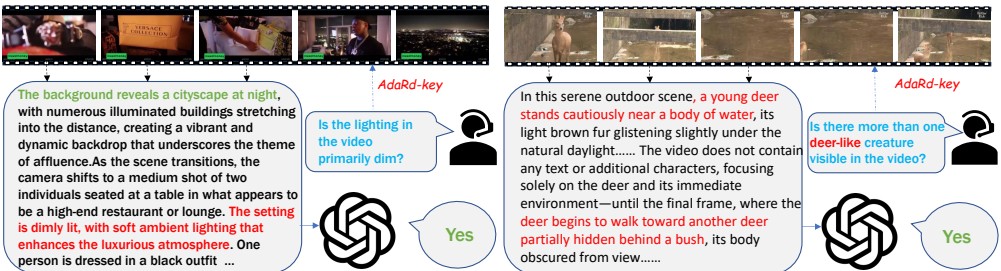

Figure 6: AdaRD-Key for video captioning with Qwen2.5-VL, evaluated by an external judge. For each clip, AdaRD-Key first selects keyframes (dotted arrows over the filmstrips) and Qwen2.5-VL generates a caption from these frames; red phrases in the caption highlight evidence. We then query Gemini-2.5-Flash-Lite to decide whether the caption alone is sufficient to answer the question shown in blue, and report the verdict on the right. In both examples, captions produced from AdaRD-Key frames contain the necessary information, demonstrating that our sampler supplies question-relevant context across the video.

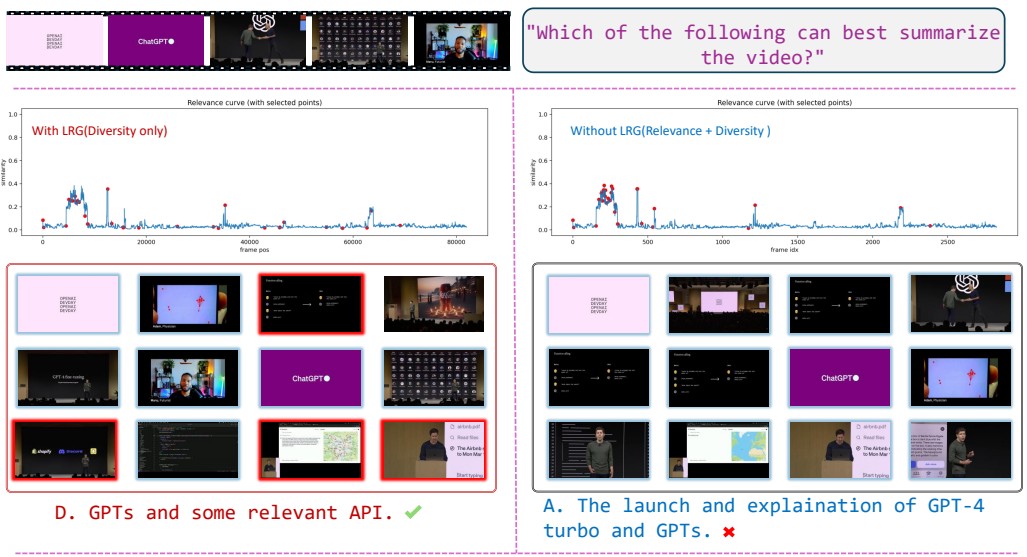

Figure 7: Impact of the Lightweight Relevance-Aware Gate (LRG) under weak query–video alignment using Qwen2-VL with 32 frames.

