# OpenReview forum: "AdaRD-Key: Adaptive Relevance-Diversity Keyframe Sampling for Long-form Video Understanding"
_ICLR.cc/2026/Conference — ICLR 2026 Conference Withdrawn Submission_

### Official Review · Reviewer_xNLc · 2025-10-14

**Soundness:** 3
**Presentation:** 3
**Contribution:** 2
**Rating:** 4
**Confidence:** 4

**Summary:**

This paper proposes AdaRD-Key, a keyframe sampling framework for query-driven long-form video understanding. The method introduces a joint Relevance–Diversity Max-Volume (RD-MV) objective that simultaneously encourages semantic relevance and visual diversity during keyframe selection. Two auxiliary modules enhance adaptability: Variability–Budget Scaling (VB-Scale) automatically adjusts the trade-off coefficient $\lambda$ according to the variability of query–frame relevance scores and the video length; a lightweight relevance-aware gate detects weak query–video alignment and switches to a diversity-only objective to preserve coverage. AdaRD-Key is fully training-free, efficient on a single GPU, and applicable to existing VLMs such as LLaVA-Video and Qwen2-VL. Experiments on LongVideoBench, Video-MME, and VCapsBench show consistent performance gains over uniform sampling, AKS, and MaxInfo baselines.

**Strengths:**

- **Practical design.** The framework is training-free and compatible with various pretrained multimodal LLMs, providing immediate utility for long-video reasoning pipelines. The method achieves near-linear computational cost and real-time inference on a single GPU, which is valuable for large-scale video datasets.
- **Technical soundness.** The RD-MV objective is mathematically well-defined and optimized efficiently through greedy submodular maximization with Sherman–Morrison updates.
- **Clear modular analysis.** The ablation study transparently evaluates the contributions of relevance, diversity, VB-Scale, and gating, confirming the incremental benefits of each.

**Weaknesses:**

- **Marginal improvement over baselines.** Although AdaRD-Key slightly outperforms prior training-free methods (e.g., AKS, MaxInfo), the gains are small and within variance. The paper lacks comparisons with adaptive temporal search methods (e.g., _LV-Haystack (T*)_, _Videotree_) and an analysis of complexity–efficiency trade-offs to justify the added design overhead.
- **Heuristic parameterization.** Key parameters—such as the gating threshold $\tau {=},0.4$ and the $\lambda$ bounds in VB-Scale—are heuristic with no sensitivity analysis. The VB-Scale mechanism itself lacks theoretical grounding or quantitative evaluation, and reliance on BLIP-2 relevance scores may reduce adaptability.
- **Lack of qualitative error analysis.** Results are reported only as averaged accuracies without qualitative or failure analyses. Including more diverse datasets would strengthen generalization evidence.
- **Presentation clarity.** The related-work section reads as a citation list rather than a structured taxonomy, and the paper does not clearly justify why combining relevance with $\log\det$-based diversity provides new insights beyond existing methods like MaxInfo or AKS.
- **Limited evaluation coverage.** The evaluation covers LongVideoBench and VideoMME, but omits some well-known long-video benchmarks such as MLVU, LVBench.
- **Incomplete coverage of related work.** The discussion omits several recent and directly relevant studies on keyframe selection and long-video reasoning, such as _Logic-in-Frames_, _LV-Haystack (T*)_, _Videotree_ (CVPR 2025) ... Without contextualizing against these advances, the novelty appears incremental.

**Questions:**

- Accuracy suffices for baseline comparison, but **additional metrics** reflecting frame diversity, redundancy, or query–frame alignment would better validate the proposed RD-MV objective.
- Expand the **literature review** to cover recent 2024–2025 works in adaptive keyframe selection, temporal retrieval, and long-video reasoning. Discuss how AdaRD-Key differs from these temporal search and related retrieval-driven approaches.
- A **sensitivity analysis** of $\lambda$, $\tau$, and the diversity budget $\rho$—along with qualitative visualizations of both successful and failed selections—would strengthen the empirical evidence. How stable are the results across different parameter values?
- An explicit analysis of **computational complexity** and **runtime scaling** would help justify the added design components of AdaRD-Key relative to its modest accuracy gains.
- Could query complexity or confidence be used to **dynamically adjust** the number of selected frames rather than fixing $K$?
- How does the **diversity-only fallback** behave in videos containing **multiple semantically relevant events**? Are there cases where this fallback leads to missing query-relevant evidence?
- Are there **qualitative examples** where AdaRD-Key **fails** under noisy, ambiguous, or semantically diffuse queries? Such an analysis would clarify the model’s practical limitations.

_If the above concerns are addressed, I would be inclined to raise my score._

---

### Official Review · Reviewer_9C1o · 2025-10-21

**Soundness:** 2
**Presentation:** 1
**Contribution:** 2
**Rating:** 2
**Confidence:** 5

**Summary:**

The paper introduces AdaRD-Key, a training-free and plug-and-play method for selecting informative keyframes from long videos to support downstream vision–language tasks such as video question answering and captioning. Unlike uniform or purely relevance-based sampling, AdaRD-Key dynamically balances query relevance and visual diversity in a query-adaptive manner. Its key innovations include a mechanism that adjusts the trade-off between relevance and diversity based on video duration and the variability of frame-level relevance scores, as well as a lightweight gating strategy that gracefully degrades to diversity-only selection when the input query is weakly aligned with the video content. Evaluated on benchmarks like LongVideoBench, Video-MME, and VCapsBench, AdaRD-Key consistently outperforms existing sampling strategies—especially on longer videos, while requiring no model training or fine-tuning.

**Strengths:**

The paper’s primary strength lies in its practical and well-engineered system design. It effectively combines established ideas, submodular diversity (via log-determinant), query–frame relevance scoring, and adaptive weighting into a cohesive, training-free pipeline tailored for long-video understanding. The quality of execution is solid: the method is computationally efficient, includes sensible heuristics, and is evaluated consistently across multiple benchmarks with clear ablations. The clarity of presentation is strong, with intuitive explanations of design choices and qualitative results.

**Weaknesses:**

- The manuscript would benefit from careful revision. Some typos and mistakes in the article are as follows：
  - Can $k$ in line 148 and Equation (1) and $K$ in line 200 and line 203 be represented by the same symbol? Maybe this applies to the entire article.
  -  $\mathcal{D}(F)$ in Equation (2) should be $D(F)$ ?
  - Inconsistent citation format with Introduction in Line 215 and 231 `BLIP-2 Li et al. (2023)`
  - Line 227 has an unknown meaning of `14This`
  - The budget ratio is repeatedly defined on line 200,  $\rho = N / K$, and on line 207, $ \rho = T / K$. Not sure about the difference

- The use of formulas and symbols in the article lacks academic rigor.
  - $R(f)$ means a function with relevance scores in the `section 3.2.2`. However, $R$ becomes a specific relevance score in the `section 3.2.3` and  Equation (10). This inconsistency and unclear image reader's understanding.
  - Equation (10) uses CV directly as the variable name in formula 10, which is not academic enough.

- The article claims AdaRD-Key employs a lightweight relevance-aware gating mechanism. However, the necessary analysis of latency and computational cost is currently lacking.

**Questions:**

- I want to know the systematic experiments and analysis of the latency and computational cost, especially the consumption of 1fps candidate frames for long videos.

- The AdaRD-Key introduces some hyperparameters, including the threshold $\tau$, $\lambda_{min}$, $\lambda_{max}$, $\alpha$, $\rho$ and $\rho_{0}$. It is unclear whether the choice of these hyperparameters was based on experience or experimentation. If it is based on experiments, it is necessary to supplement the corresponding ablation experiments.

- The proposed AdaRD-Key conducted experiments on Video-MME and LongVideoBench, lacking experimental results on another widely used MLVU benchmark [1].

- The paper chooses BLIP-2 for encoding, while similar methods [2,3 ] use the CLIP-series of models. I am curious about the considerations behind this choice.

[1] Zhou J, Shu Y, Zhao B, et al. MLVU: A comprehensive benchmark for multi-task long video understanding[J]. arXiv e-prints, 2024: arXiv: 2406.04264.

[2] Tang X, Qiu J, Xie L, et al. Adaptive keyframe sampling for long video understanding[C]//Proceedings of the Computer Vision and Pattern Recognition Conference. 2025: 29118-29128.

[3] Zhang S, Yang J, Yin J, et al. Q-Frame: Query-aware Frame Selection and Multi-Resolution Adaptation for Video-LLMs[J]. arXiv preprint arXiv:2506.22139, 2025.

---

### Official Review · Reviewer_Nygr · 2025-10-29

**Soundness:** 2
**Presentation:** 2
**Contribution:** 2
**Rating:** 4
**Confidence:** 4

**Summary:**

The paper proposes AdaRD-Key, a training-free keyframe selector for long video understanding. It maximizes a joint objective that adds a query-conditioned relevance term to a log-determinant diversity term, and uses a greedy Max-Volume selector with Sherman–Morrison updates for efficiency. A lightweight relevance-aware gate switches to diversity-only mode when query–video alignment is weak, and a variability–budget scaling adapts the relevance–diversity trade-off to the score distribution and frame budget. The method is plug-and-play for existing VLMs, and reports gains on LongVideoBench and Video-MME under 32/64-frame budgets, with ablations for each component.

**Strengths:**

The motivation is clear.
The method is training-free and simple to deploy, with a single objective that couples relevance and diversity.
The paper presents results on two long-video benchmarks and shows consistent improvements at fixed budgets, along with ablations that incrementally add relevance, diversity, gating, and variability–budget scaling.

**Weaknesses:**

First, model scale is limited: all main results use 7B VLMs. The absolute gains are modest and there is no evidence that improvements hold for larger models, so it is hard to tell whether the method truly increases information quality rather than exploiting small-model sensitivity.

Second, the comparisons keep the same frame budget (for example 32 or 64 frames) across methods. This shows “better selection at the same budget,” but does not demonstrate the compression advantage of frame searching (for example matching a 64-frame baseline with far fewer frames).

Third, there is no direct measurement of frame search quality; all evaluation is indirect through downstream VQA. A dedicated retrieval-style metric or use of datasets designed to test temporal search (for example in the spirit of “LongVideoHaystack”) would isolate selection quality from VLM confounds.

**Questions:**

Can you report results with larger VLMs to check whether gains persist or grow with scale?

Can you demonstrate compression benefits, for example matching a 64-frame baseline using 8 frames with your selector?

Can you add a direct frame-search quality evaluation (for example retrieval/recall of query-evidence segments), in addition to downstream accuracy?

---

### Official Review · Reviewer_T4xh · 2025-10-31

**Soundness:** 2
**Presentation:** 3
**Contribution:** 2
**Rating:** 4
**Confidence:** 3

**Summary:**

This paper presents a key frame selection for video understanding. They introduce a RD-MV objects to maximum the retrieved video quality for VQA task.  The framework is plug-and-play, requires no additional training. They carry experiment on LongVideoBench, Video-MME and VCapsBench.

**Strengths:**

1. The paper is well-written and easy to follow.
2. The approach is simple and training-free.
3. The evaluation is comprehensive on three datasets.

**Weaknesses:**

a, the improvement over previous methods is small.
b, limited novelty: Query conditioned retrieval+max info does not appear novel (AKS+MAXinfo)
c, lack of evidence to support the real-time claim.
d, the work claims the redundancy reduction to previous methods. Evidence in comparison is not provided.

**Questions:**

1, Why does the author choose BLIP as the relevance encoder, instead of other methods (CLIP, Llava,Qwen)
2, What is the additional cost of the module?
3, How would this work behave in scenarios that queries does not include specific item and locating the key frame is difficult by Bilp similarity.  For example in Fig 3, what is the 40cm is not displayed in the video.
4, what is the limitation of this work?
5, The improvement to AKS is very marginal, could this be experimental randomness?

---

### Note · Authors · 2025-11-26

I have read and agree with the venue's withdrawal policy on behalf of myself and my co-authors.